# Crystal Growth of Osmium(IV) Dioxide in Chlorine-Bearing Hydrothermal Fluids

Haibo Yan [1,2,3], Zhuoyu Liu [1], Jian Di [1,3,4] and Xing Ding [1,3,*]

1 State Key Laboratory of Isotope Geochemistry, Guangzhou Institute of Geochemistry, Chinese Academy of Sciences, Guangzhou 510640, China
2 CAS Key Laboratory of Mineralogy and Metallogeny, Guangzhou Institute of Geochemistry, Chinese Academy of Sciences, Guangzhou 510640, China
3 CAS Center for Excellence in Deep Earth Science, Guangzhou 510640, China
4 College of Earth and Planetary Sciences, University of Chinese Academy of Sciences, Beijing 100049, China
* Correspondence: xding@gig.ac.cn

**Abstract:** A mineral's morphology is usually related to its growth process and environment. This study reported crystal growth of $OsO_2$ through hydrolysis experiments of $K_2OsCl_6$ at 150–550 °C and 100 MPa to investigate the growth mechanism of $OsO_2$ and the transport and enrichment of Os in chlorine-bearing hydrothermal fluids. Time-series experimental results showed that the $OsO_2$ crystals grow from 40–150 nm irregular nanoparticles to 150–450 nm nanospheres with time. As the temperature and initial solution concentrations increase, $OsO_2$ can form more uniform and larger $OsO_2$ nanosphere crystals, suggesting a positive effect of temperature and initial solution concentration on the crystal growth of $OsO_2$. The results indicate that the nucleation and aggregate growth driven by the hydrolysis of Os–chloride complex controls the early growth of $OsO_2$ crystals for a short duration; however, after the hydrolysis reaches equilibrium, the growth process of $OsO_2$ nanosphere crystals is dominated mostly by the Ostwald ripening, where the diffusion of Os ions along the fluid–nanocrystal boundary facilitates the coarsening. Given that the transport and cycle of Os from the lithosphere to the hydrosphere is controlled mainly by the stability of the Os–chloride complex, $OsO_2$ nanosphere crystals could occur in seafloor hydrothermal vent systems.

**Keywords:** osmium; crystal growth; nanosphere; chlorine-bearing fluids; ostwald ripening

## 1. Introduction

Water–rock interactions are one of the most critical mineralization and material recycling processes on earth [1]. They can occur widely in seafloor hydrothermal vent systems, promoting metal extraction from the mantle-derived magma and wall rocks [2–6]. In such a scenario, the extracted metal elements can be transported in the hydrothermal fluids and enriched in the minerals, encrustations, and sediments, resulting in a potentially substantial resource of minerals such as Fe, Mn, Zn, Pb, Co, Cu, Ag, Sn, REEs, PGEs, etc. [7–10]. Among them, PGEs usually transport as chloride complexes in the deep sea and are then enriched in the encrustations as isomorphic compounds or metals or are scattered in the ocean floor sediments, providing substantial potential resources [11,12]. In seafloor hydrothermal systems, PGEs are usually oxidized to a high state (+2 or +4) by oxidized seafloor hydrothermal fluids and then precipitated into the ocean sediments and muds [13,14], contrasting sharply with the low state (0 and +2) PGEs which could exist during the magmatic process in the deep earth [15–17].

As a representative element of PGEs, osmium (Os) is an important functional and structural material with high ionic and magnetic conductivity as well as high temperature-resistant and corrosion-resistant properties [18]. It can be widely used in microbial reagents, radiation protection materials, catalysts, gas fixing agents, electrode conductivity materials, and other fields [19–22]. Osmium-rich crystal materials such as osmium (IV) dioxide

also have an essential role in electrochemical and electro-mechanical industries [21,23]. However, only a rutile-type $OsO_2$ has been synthesized using the chemical vapor transport (CVT) method and researched [18,24]. Other precipitation and nucleation properties of high-valence Os ions, as well as their crystal growth in hydrothermal fluids, are still poorly understood, which hinders future applications and a deep understanding of Os enrichment and deposition in the oxidized seafloor hydrothermal fluids.

To fill this gap, this study adopts experimental methods to probe the precipitation process of Os(IV) and its growth mechanism in chlorine-bearing hydrothermal fluids. This study could enrich our understanding of the transport and enrichment of Os in the hydrothermal vent system.

## 2. Materials and Methods

### 2.1. Materials and Instruments

The experimental materials in this study included potassium chlorosmate ($K_2OsCl_6$, Os $\geq$ 38.7%, AR, Macklin, Shanghai, China) and ethanol (AR, Huada, Guangzhou, China). $K_2OsCl_6$ has a high density of 3.470 g/cm$^3$ and is one of the most stable chemicals among Os–Cl complexes at room temperature and ambient pressure. It was diluted with deionized water to produce the initial experimental solutions with different concentrations of 0.002 and 0.005 mol/L. Note that all operations related to highly corrosive and toxic reagents should be carried out in a fume hood and the operators must be protected by wearing protective equipment during the whole experimental process.

The experiments were performed using Tuttle-type cold-seal pressure vessels with a 27 mm outer diameter, a 6 mm inner diameter, and a 250 mm length, which is suitable for high temperature–pressure experiments up to 950 °C and 500 MPa at the hydrothermal laboratory of the Guangzhou Institute of Geochemistry, China [25–29]. The pressure was transmitted by the deionized water and monitored with a high-precision high-pressure gauge ($\pm$5 MPa). The temperature was controlled by the heating furnace and measured with a high-accuracy electronic temperature controller ($\pm$5 °C) using a NiCr-Ni (K-type) thermocouple. A constant-temperature drying oven ($\pm$1 °C, DGG-9070B, Jiangdong, China) was used to process the experimental products.

The analysis equipment used in this study included a field-emission scanning electron microscope (SEM) equipped with an energy-dispersive spectrometer (EDS), a high-resolution confocal Raman spectrometer, and a micro-area X-ray Diffraction (μ-XRD). The field-emission scanning electron microscope (SEM) (SU8010, HITACHI, Tokyo, Japan) and an EDAX Apollo x-SDD energy dispersive spectrometer (EDS) (Core Lab, Tulsa, OK, USA) were used for morphological and composition analysis of the crystals with a working voltage of 15 kV and a magnification of 600–80,000 at the State Key Laboratory of Organic Geochemistry of the Guangzhou Institute of Geochemistry, China [26]. The high-resolution confocal Raman spectrometer (alpha 300R, WITec Instruments Corp, Ulm, Germany) was equipped with three lasers (488, 532, and 633 nm), three gratings (300, 600, and 1800 grooves/mm), and four Zeiss objectives (5$\times$, 20$\times$, 50$\times$, and 100$\times$) and was used for the qualitative analysis of the crystals with a laser power of 10 mW, an integration time of 6 s, and a laser wavelength of 532 nm at the hydrothermal laboratory of Guangzhou Institute of Geochemistry, China [30]. A Dmax RAPID V micro-area X-ray Diffraction (μ-XRD) (Rigaku, Tokyo, Japan) with a working voltage of 40 kV, working current of 30 mA, exposure time of 100 s, and collimator of 0.1 mm was used for the structure analysis of the crystals at the CAS Key Laboratory of Mineralogy and Metallogeny of the Guangzhou Institute of Geochemistry, China.

### 2.2. Experimental Process

Firstly, gold capsules with a diameter of 5 mm and a length of 20–30 mm were prepared by acid boiling, quenching, and cleaning with deionized water and ethanol and was used as the sample reactant container. Then, the container was filled with the initial experimental solutions accounting for 40–60% of the volume and sealed at both ends with a tungsten

inert gas welding system (PUK U3, Germany). Next, the sample reactant container was placed into the Tuttle-type cold-seal pressure vessels and followed by a nickel filler rod to prevent the sample reactant container from bobbing due to heating. After the experiments were finished, the vessels were quickly quenched in an ice–water mixture to drop the temperature to below 100 °C in several seconds. Finally, the sample reactant container was taken out from the cold-seal pressure vessels and then unfolded, cleaned repeatedly with deionized water and alcohol, and dried for analysis. Note that before and after all experiments, the capsule was checked for leaks by weighing and heating.

## 3. Results and Discussion

$K_2OsCl_6$ can remain stable at room temperature and ambient pressure but gradually hydrolyzes and eventually forms a precipitate at high temperature and high pressure [31]. The cumulative hydrolysis reaction and formation of $OsO_2$ crystal can be expressed as:

$$K_2OsCl_6(aq) + 2H_2O \rightarrow OsO_2(s) + 4HCl(aq) + 2\ KCl(aq) \tag{1}$$

The above reaction can occur rapidly and precipitate within a few hours, producing $OsO_2$ crystals, hydrochloric acid, and chloride ions to create an acidic and chlorine-bearing environment [27]. All experiments were performed at a pressure of 100 MPa to avoid the effect of pressure on the $OsO_2$ crystals. The experimental products were identified as $OsO_2$ precipitates by the EDS (Figure 1) and the Raman spectrum (Figure 2). Moreover, the XRD spectra also showed that the crystals at 150 and 550 °C, 24 h, and 100 MPa using the 0.005 mol/L initial solution were $OsO_2$, indicating that the experimental products from 150 to 550 °C were $OsO_2$ crystals (Figure 3). All the identifications indicated that the experimental products were $OsO_2$ crystals. The morphology of the $OsO_2$ precipitates was analyzed with the SEM, which clearly shows that the $OsO_2$ crystals were in the form of nanospheres or irregular nanoparticles with a diameter of 40–500 nm at different magnifications (Figures 4 and 5). All experimental conditions and the results of $OsO_2$ crystals are shown in Table 1.

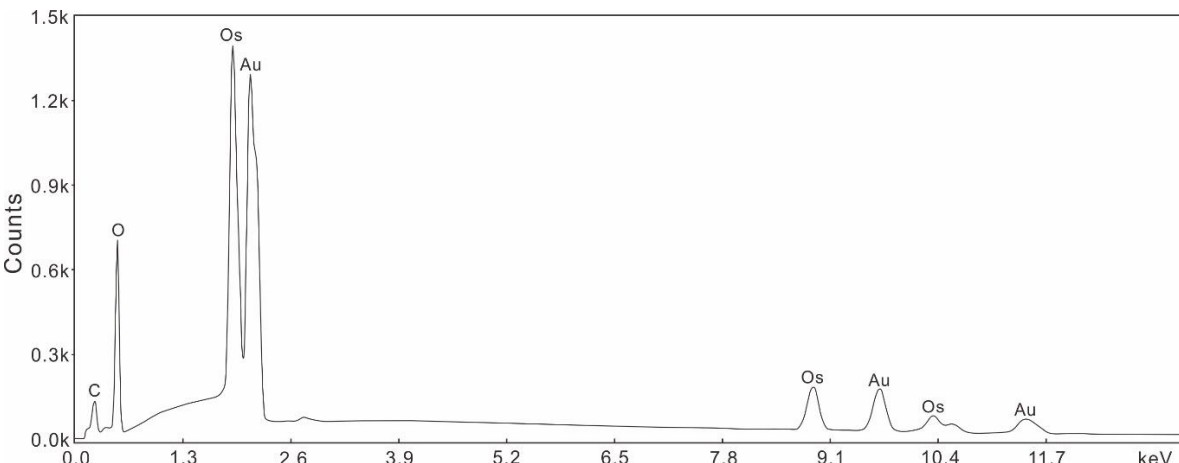

**Figure 1.** Representative EDS analysis of $OsO_2$ adhered to the inner surface of the gold capsule. Note that Au is derived from the gold capsule.

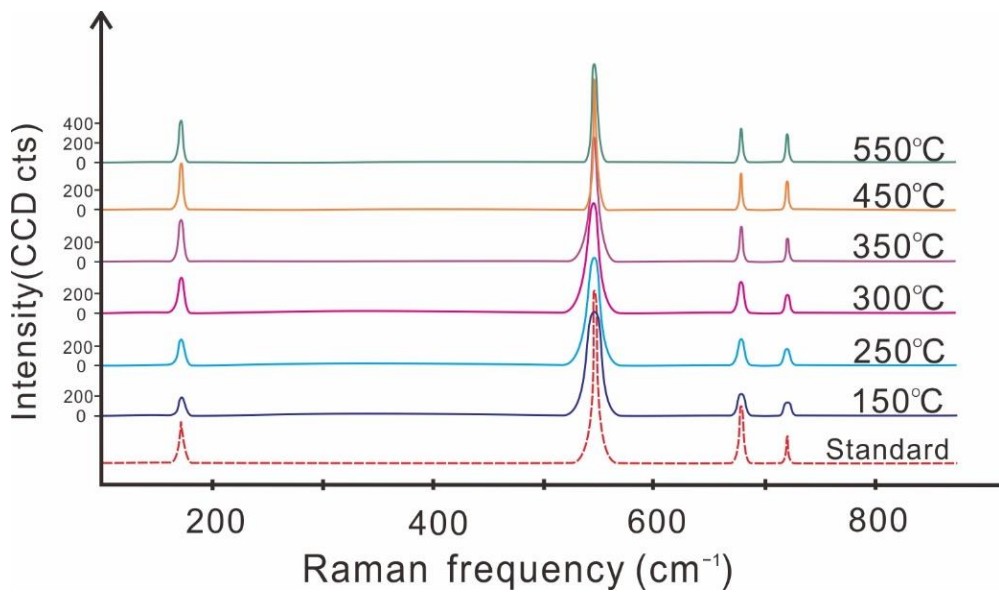

**Figure 2.** Raman spectra of OsO$_2$ in the present study and typical OsO$_2$ crystals. The standard Raman spectrum of OsO$_2$ is derived from Yen, et al. [32].

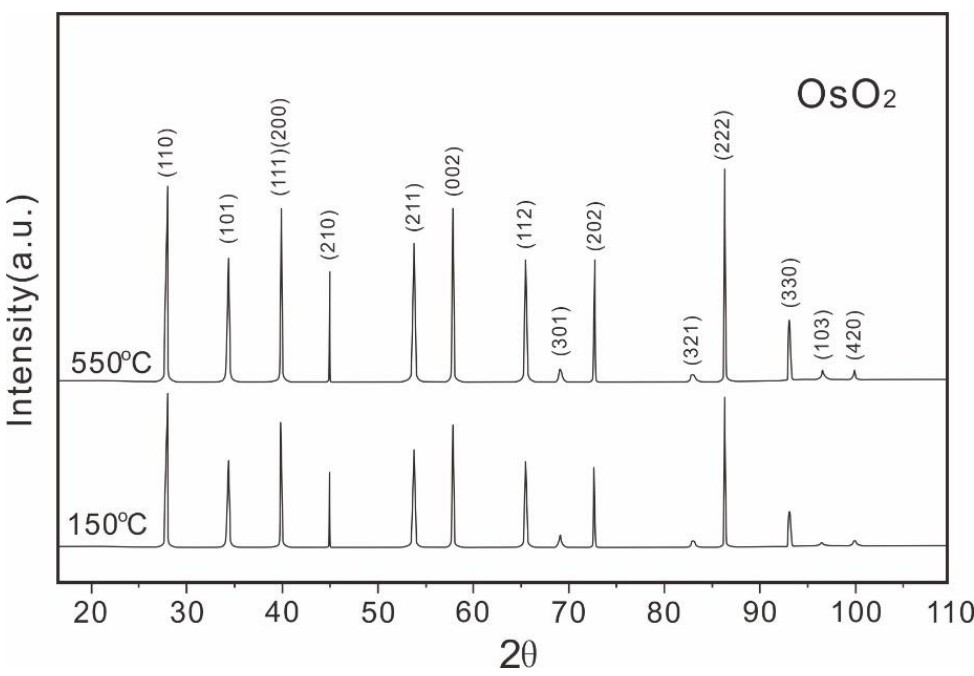

**Figure 3.** Representative XRD analysis of the OsO$_2$ crystals at 150 and 550 °C, 24 h, and 100 MPa using 0.005 mol/L initial solutions.

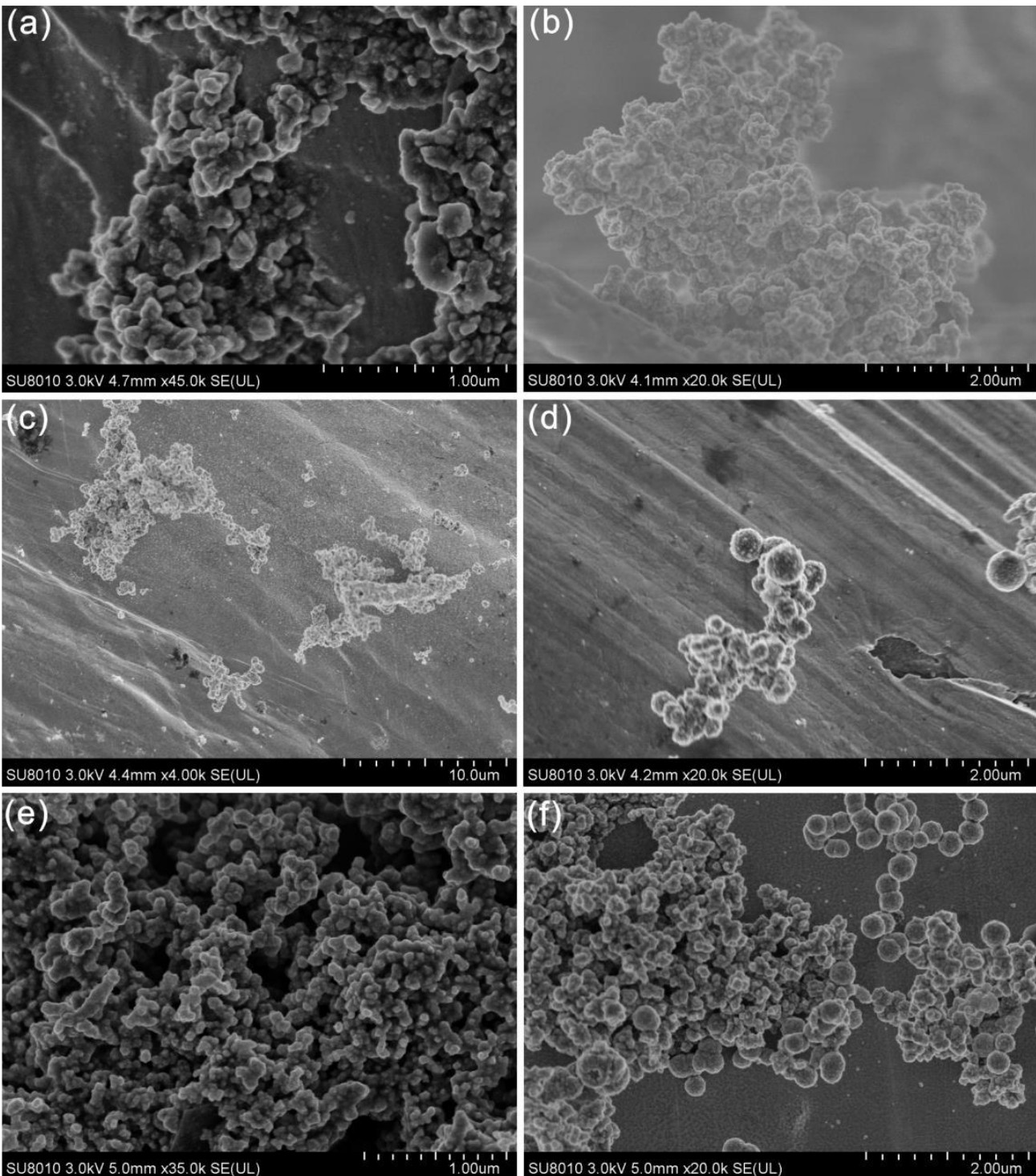

**Figure 4.** Representative SEM micrographs of $OsO_2$ nanospheres using 0.002 mol/L initial solution for: (**a**) No. 1 at 300 °C, 100 MPa, and 5 h; (**b**) No. 2 at 300 °C, 100 MPa, and 12 h; (**c**) No. 3 at 300 °C, 100 MPa, and 24 h; (**d**) No. 4 at 300 °C, 100 MPa, and 36 h; (**e**) No. 5 at 250 °C, 100 MPa, and 24 h; and (**f**) No. 6 at 450 °C, 100 MPa, and 24 h.

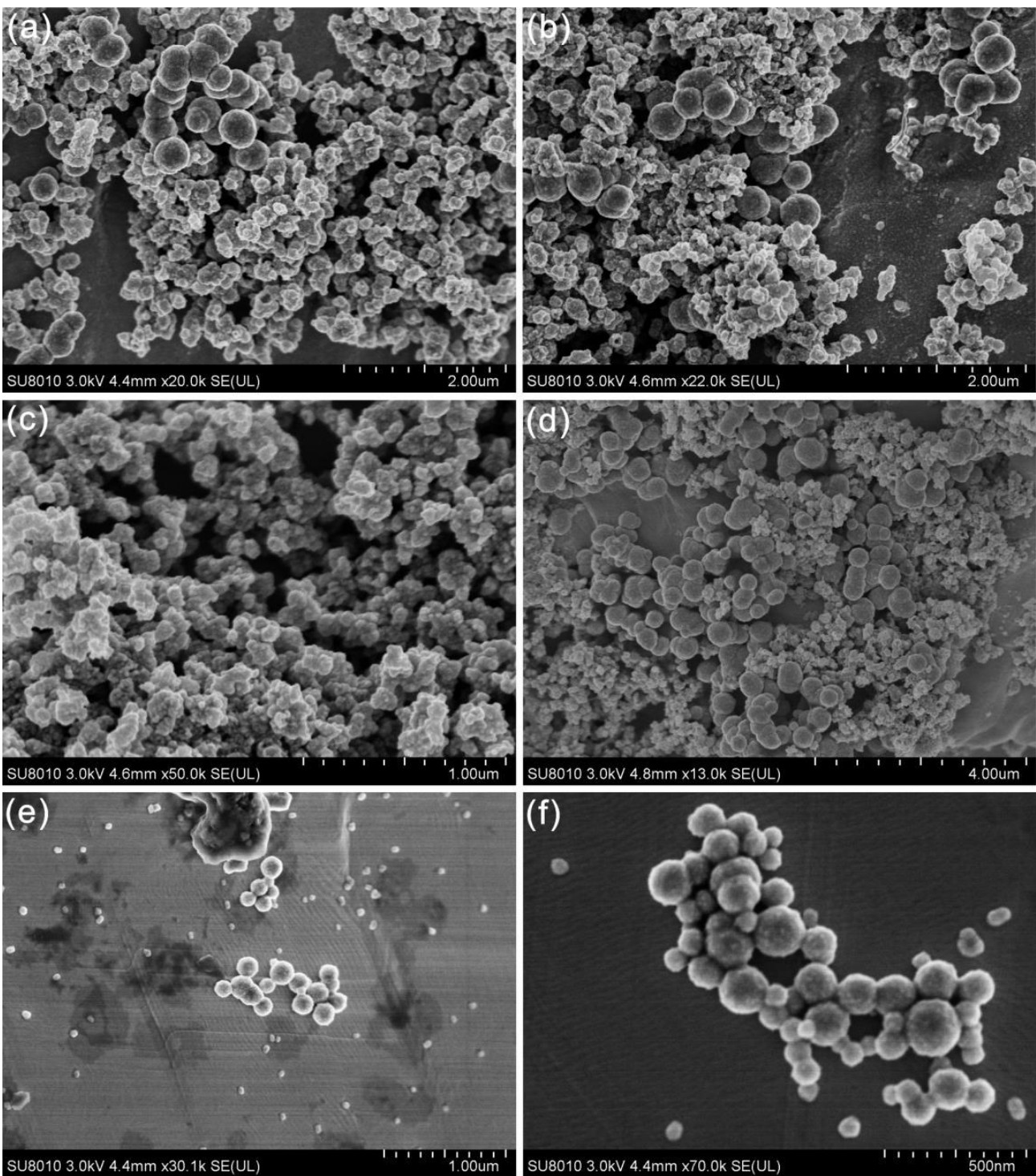

**Figure 5.** Representative SEM micrographs of OsO$_2$ nanospheres using 0.005 mol/L initial solution for: (**a**) No. 7 at 150 °C, 100 MPa, and 24 h; (**b**) No. 8 at 250 °C, 100 MPa, and 24 h; (**c**) No. 9 at 350 °C, 100 MPa, and 24 h; (**d**) No. 10 at 450 °C, 100 MPa, and 24 h; and (**e,f**) No. 11 at 550 °C, 100 MPa, and 24 h.

**Table 1.** Preparation of the OsO$_2$ nanospheres under various conditions.

| No. | Initial K$_2$OsCl$_6$ Concentration (mol/L) | Temperature (°C) | Pressure (MPa) | Time (h) | Diameter (nm) |
|-----|---------------------------------------------|------------------|----------------|----------|---------------|
| 1 | 0.002 | 300 | 100 | 5 | 40–150 |
| 2 | 0.002 | 300 | 100 | 12 | 100–280 |
| 3 | 0.002 | 300 | 100 | 24 | 100–380 |
| 4 | 0.002 | 300 | 100 | 36 | 150–450 |
| 5 | 0.002 | 250 | 100 | 24 | 50–150 |
| 6 | 0.002 | 450 | 100 | 24 | 100–400 |
| 7 | 0.005 | 150 | 100 | 24 | 50–400 |
| 8 | 0.005 | 250 | 100 | 24 | 100–400 |
| 9 | 0.005 | 350 | 100 | 24 | 50–200 |
| 10 | 0.005 | 450 | 100 | 24 | 100–500 |
| 11 | 0.005 | 550 | 100 | 24 | 50–300 |

*3.1. Time Effect on the OsO$_2$ Crystals*

Time series experiments showed that the hydrolysis time provides direct constraints on the crystal structure and size of OsO$_2$ precipitates. As shown in Figure 4a–d, at 300 °C, 100 MPa, and 0.002 mol/L initial solution, the OsO$_2$ crystals were in the form of irregular nanoparticles and grew from the diameter of 40–150 nm to the diameter of 100–280 nm when the time was increased from 5 to 12 h. When the time was increased from 12 to 24 h, the OsO$_2$ crystals grew from irregular nanoparticles with a diameter of 100–280 nm to nanosphere crystals with 100–380 nm in size. As time was elevated from 24 to 36 h, the OsO$_2$ crystals increased in size from 100–380 nm to 150–450 nm, presenting as nanosphere OsO$_2$ crystals. These data indicated that the OsO$_2$ crystals can grow from irregular nuclei to nanospheres with time. The OsO$_2$ crystals tended to be complete nanosphere crystals after 24 h, as the hydrolysis reaction can reach equilibrium within 24 h, according to previous studies [25,27]. Hence, all following experiments were performed for 24 h to ensure the integrity of the OsO$_2$ nanosphere crystals.

Given that the size of the OsO$_2$ crystals could be increased with time at similar conditions of 300 °C, 100 MPa, and 0.002 mol/L initial solution concentration, the growth rate of the OsO$_2$ crystals could be estimated by counting the sizes of OsO$_2$ crystals at various timepoints, as shown in Figure 6. The results showed that the growth rate of OsO$_2$ crystals reached 18.6 nm/h before 12 h, dropped to 8.3 nm/h from 12–24 h, and was only 5.8 nm/h after 24 h, which can be mainly attributed to the hydrolysis rate of K$_2$OsCl$_6$. The hydrolysis rate of K$_2$OsCl$_6$ reaches a maximum value within 12 h because of the promotion of a high initial K$_2$OsCl$_6$ concentration for reaction (1), resulting in the rapid growth of OsO$_2$ crystals within the first 12 h. Then, the hydrolysis rate decreases after 12 h due to the decreasing initial Os concentration and the elevated concentrations of HCl and KCl in the aqueous solution, which reduces the reaction (1) rate and causes a decrease in the growth rate of OsO$_2$ crystals after 12 h. Finally, after the hydrolysis reaction reaches an equilibrium, the Os concentration in the aqueous solution remains stable, meaning that there exists a dynamic equilibrium between the Os concentration in the aqueous solution and the precipitated OsO$_2$, which further reduces the growth rate of OsO$_2$ crystals [29,33,34]. Accordingly, the morphology results at various timepoints suggested that the OsO$_2$ crystals gradually precipitated from the solution and clumped together to form irregular crystals before the equilibrium of the experimental reaction due to the sudden precipitation of a large amount of OsO$_2$ products in a short period. However, when the hydrolysis reaction reached equilibrium, OsO$_2$ was in the form of a complete nanosphere crystal and grew slowly in size with increasing time.

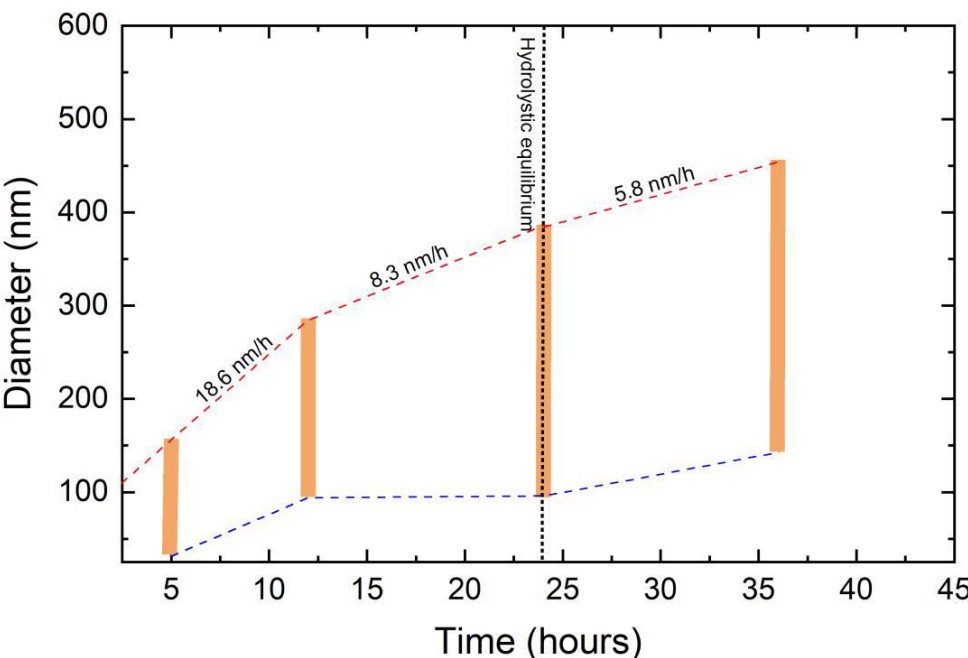

**Figure 6.** Grain size diagram of OsO$_2$ nanospheres from various timepoints synthesized at 300 °C and 100 MPa using 0.002 mol/L initial solution.

### 3.2. Temperature Dependence on the OsO$_2$ Crystals

The experimental temperature significantly affects the growth and morphology of OsO$_2$ crystals, based on the morphology results from SEM. As expounded by previous studies, the temperature can affect the degree of the hydrolysis reaction and thus restrict the morphology of hydrolysis products [26,35,36]. In this study, the temperature presented an obvious restriction on the morphology of OsO$_2$. The OsO$_2$ grew from 50–150 nm to 100–380 nm to 100–400 nm and was more uniform with temperature increasing from 250 °C to 300 °C to 450 °C at low initial concentrations solution of 0.002 mol/L and 100 MPa (Figure 4c,e–f). As shown in Figure 5, at the high initial concentration of 0.005 mol/L initial solution, 100 MPa, and 24 h, the OsO$_2$ was in either the form of small nanospheres or irregular nanospheres with a diameter of 50–400 nm at 150 °C (Figure 5a) and a diameter of 100–400 nm at 250 °C (Figure 5b). OsO$_2$ presented as 50–200 nm nanosphere crystals at 350 °C (Figure 5c) but became more uniform and complete nanospheres with sizes of 100–500 nm at 450 °C (Figure 5d) and 50–300 nm at 550 °C (Figure 5e,f). As the result of the morphology of the OsO$_2$ crystals at varying temperatures, the crystal structure of OsO$_2$ became more uniform and complete when the temperature increased from 150 to 550 °C.

Building upon the statistics for the size of the OsO$_2$ nanosphere crystals at different temperatures, the size of OsO$_2$ crystals as a whole showed a gradual increase from 160–185 to 215–240 nm when temperatures were elevated from 150 to 550 °C, indicating that elevated temperature has a positive effect on the size of the OsO$_2$ crystals (Figure 7). The results show that the OsO$_2$ crystals gradually increased in size from 160–185 to 185–210, 215–240, and 215–240 nm when temperatures were elevated from 150 to 250, 450, and 550 °C, respectively. This shows that temperature has a promoting effect on the growth of the OsO$_2$ crystals. However, the OsO$_2$ crystals at 350 °C had a more uniform size but smaller diameter than that of other temperatures, for which a possible slight oscillation of the experimental process and the difference in the statistical region may be responsible. Hence, the effect of temperature on the morphology of OsO$_2$ nanosphere crystals can be explained by the hydrolysis process of K$_2$OsCl$_6$. Elevated temperature can promote reaction (1) and increase the reaction rate of ions and molecules, further resulting in a large amount of OsO$_2$ being precipitated from the solution and the growth of OsO$_2$ crystals in size [37].

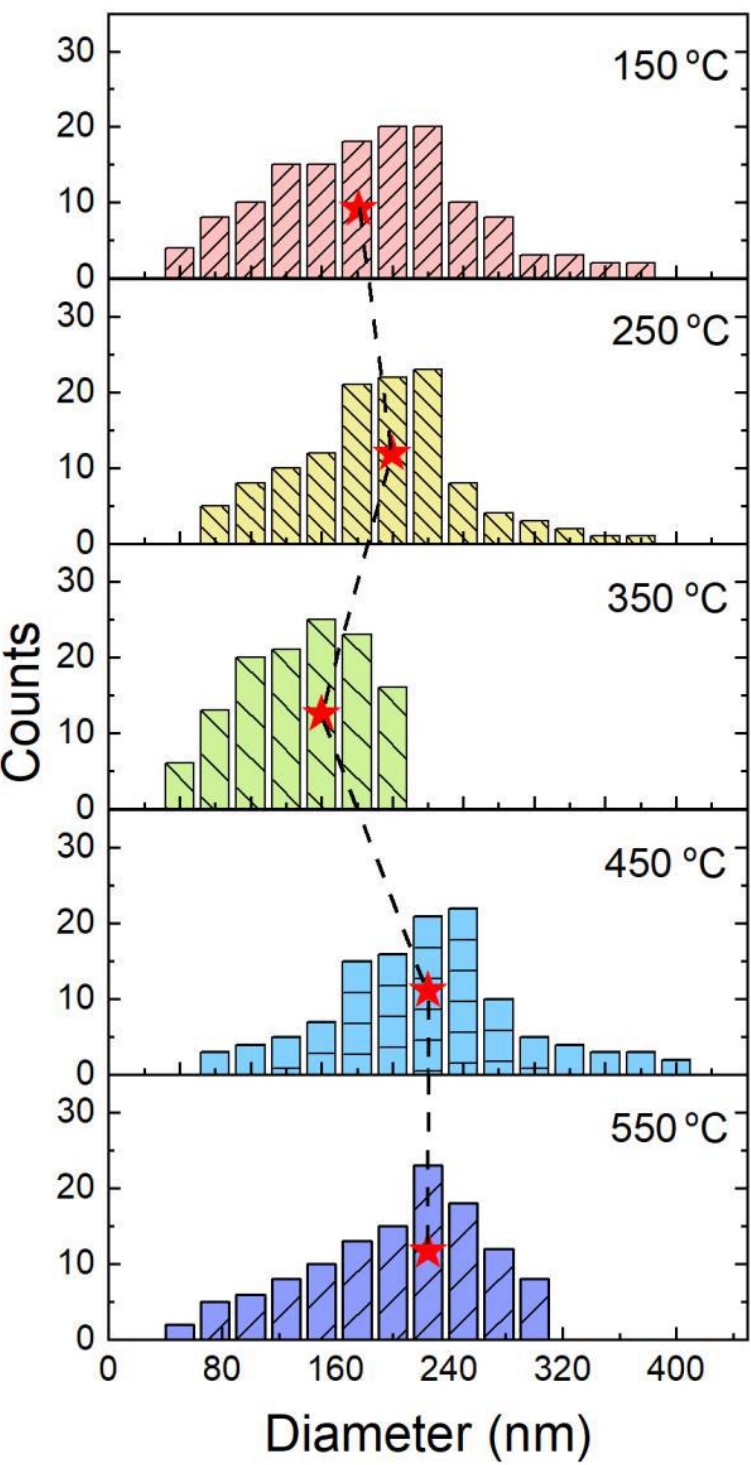

**Figure 7.** Statistical histograms of diameter sizes for $OsO_2$ nanospheres at 150–550 °C, 100 MPa, and 24 h in the 0.005 mol/L initial solution. Note that the asterisk represents the median of counts of different temperatures.

### 3.3. The Influence of Initial Solution Concentration on the $OsO_2$ Crystals

Besides time and temperature, the initial solution concentration also has an essential influence on the morphology of $OsO_2$ crystals. As shown in Figures 4e and 5b, at the same experimental conditions of 250 °C, 24 h, and 100 MPa, the $OsO_2$ crystals grew from 50–150 nm irregular nanospheres at 0.002 mol/L initial solution to a 100–400 nm nanospheres or irregular nanospheres at 0.005 mol/L initial solution. At the higher temper-

ature of 450 °C, the $OsO_2$ nanosphere crystals were also larger when using the 0.005 mol/L initial solution (a diameter of 100–500 nm) than those of the lower initial concentration solution of 0.002 mol/L (a diameter of 100–400 nm) (Figures 4f and 5d). These results suggest that the initial solution concentration has a positive effect on the growth of the $OsO_2$ crystals in size, which could be attributed to the promotion of high initial $K_2OsCl_6$ concentrations for the occurrence of reaction (1) and the formation of a large amount of $OsO_2$ precipitates during the process of hydrolysis.

*3.4. The Growth Mechanism of the $OsO_2$ Nanosphere Crystals*

Crystal nucleation and aggregate growth are usually complex processes involving the combination of atoms, particle aggregation, nucleation, and ordered combination of crystal nuclei [38,39]. In this study, with the experimental time increasing from 5 to 24 h before equilibrium, reaction (1) occurs and produces a large amount of $OsO_2$ precipitates. Then, the $OsO_2$ precipitates quickly aggregate and nucleate to form irregular nanoparticles and gradually grow, displaying a series of idiomorphic and other-shaped $OsO_2$ nanoparticles in the chlorine-bearing fluids (Figure 4a–c). This suggests that the growth of $OsO_2$ is similar to the formation mechanism of early pyrite and rutile crystals and follows the nucleation and aggregate growth pattern before the equilibrium of the hydrolysis reaction [33,40,41]. Therefore, the nucleation and aggregate growth pattern could mainly control the formation and growth of the $OsO_2$ crystals.

When reaction (1) is close to or reaches the equilibrium, the $OsO_2$ precipitates are in a dynamic equilibrium of dissolution and precipitation. The $OsO_2$ exist as irregular nanoparticles and nanospheres with a wide diameter of 100–450 nm and grow gradually with time (Figure 4c–f). The process can mainly be explained by Ostwald ripening, rather than the nucleation and aggregate growth of crystals [33,42–45]. Under the dynamic equilibrium of $OsO_2$ precipitates and the dissolution of chlorine in the hydrothermal fluids, the $OsO_2$ precipitates from the reactive fluids were significantly reduced. At this time, the nucleation and aggregate growth pattern had a weak effect on the growth of the $OsO_2$ crystals. Being in the dynamic equilibrium, the small $OsO_2$ nanoparticles synthesized early gradually dissolved and decreased, while the large $OsO_2$ nanospheres gradually increased and grew in this study (Figures 4 and 5). Furthermore, as shown in Figure 8, at 300 °C, 100 MPa, and 0.002 mol/L initial solution, the maximum diameter of $OsO_2$ crystals increased with time and is entirely consistent with the growth curve of Ostwald ripening. These conclusions provide direct evidence that the Ostwald ripening dominates the formation and growth of the $OsO_2$ nanospheres with time. They also showed that the $OsO_2$ crystals' growth is controlled by the diffusion of Os ions along the fluid–nanoparticle boundary, according to the fitting equation in Figure 8 [42]. The result from Figure 8 shows that the Ostwald ripening not only controls the growth of $OsO_2$ crystals on a long time scale but could also play a role in the growth of $OsO_2$ crystals before the hydrolysis reaction reaches equilibrium. Thus, Ostwald ripening dominates the growth of $OsO_2$ crystals when the hydrolysis reaction reaches equilibrium, but imparts a limited effect on the occurrence of the $OsO_2$ crystals on a short time scale. Certainly, high temperature and initial concentration push the reaction (1) to the right, form a large amount of $OsO_2$ precipitates, and produce a large amount of chlorine in the hydrothermal fluids, further promoting the Ostwald ripening of the $OsO_2$ crystals and the occurrence of $OsO_2$ nanosphere crystals (Figures 4e,f and 5). Moreover, $OsO_2$ nanosphere crystals are obtained for the first time and likely have great application potential as microbial reagents, catalysts, gas fixing agents, and electrode conductivity materials because of their high surface area and good electrode conductivity [19–22].

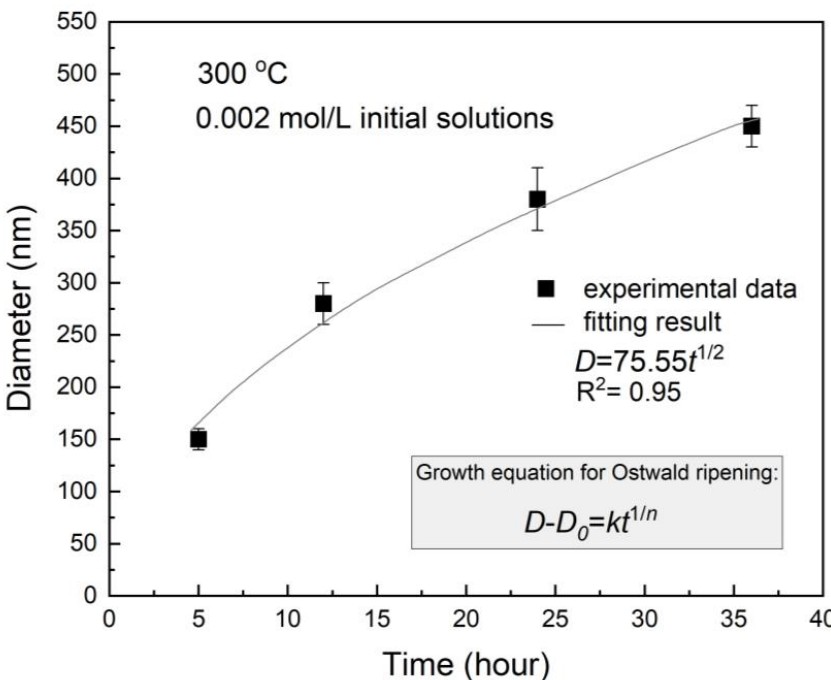

**Figure 8.** Experimental data and fitting results showing maximum diameter sizes of $OsO_2$ crystals vs. time at 300 °C and 100 MPa in 0.002 mol/L initial solutions.

### 3.5. Geological Implication

The crystal morphology of minerals is related to their growth process and environments and thus can be used as a tracer to indicate their growth environments, just as with hydrothermal pyrite, whose morphology is related to the hydrothermal temperature and properties [46,47]. Given that Os is usually stable and transported as a high coordination chloride complex, such as $OsCl_6^{2-}$, in the deep sea—though these complexes could hydrolyze and precipitate to form crystals—the morphology of $OsO_2$ crystals could shed light on the unique environment for the growth of $OsO_2$ crystals [48,49]. As mentioned in previous studies, only the rutile-type $OsO_2$ crystal has been synthesized by the chemical-vapor-transport (CVT) method and is related to the high synthesis temperature (800–940 °C) and oxygen-rich environment [18,24]. Different from the synthesis method of rutile-type $OsO_2$ crystals in the gaseous phase, the $OsO_2$ nanosphere crystals were synthesized by the hydrolysis of the Os–chloride complex for the first time in this study and could be related to the oxidized hydrothermal fluids in terms of the existence of high-valence Os. Moreover, the homogeneity of the $OsO_2$ nanosphere crystals in this study shows a stable hydrothermal fluid environment [50]. The $OsO_2$ crystals in the form of nanospheres instead of a rutile-type form in this study may be due to the dissolution of chlorine, low temperature, high pressure, and low Os saturation in hydrothermal fluids [18,48,51]. The lower temperature and higher pressure may not allow the $OsO_2$ crystal to completely grow according to a lattice structure [51]. The dissolution of chlorine and the low Os saturation also prevented the $OsO_2$ crystals from growing a rutile-type structure. In this case, the $OsO_2$ nanosphere crystals could occur in this study. Furthermore, the growth process of $OsO_2$ crystals in this study is closely related to the stability and hydrolysis of the Os–chloride complex in hydrothermal fluids and could indirectly trace the transport and cycle of Os as the Os–chloride complex from the magmatic process to the hydrothermal system [14]. Based on the growth mechanism of the $OsO_2$ crystals and the hydrolysis behavior of $K_2OsCl_6$, $OsO_2$ nanospheres could occur in relatively acidic chlorine-rich oxidized hydrothermal environments [27]. For example, they could occur in the ocean-floor sediments and muds near hydrothermal vents due to the favorable growth conditions for $OsO_2$ nanosphere crystals [7,52,53]. Accordingly, the growth of $OsO_2$ nanosphere crystals may be used as the typomorphic mineral to trace the hydrothermal conditions and fluid properties.

## 4. Conclusions

According to the hydrolysis experiments of $K_2OsCl_6$ with temperatures from 150 to 550 °C, times from 5 to 36 h, initial solution concentrations from 0.002 to 0.005 mol/L, and 100 MPa water pressure, this study obtained $OsO_2$ nanosphere crystals with a diameter of 40–500 nm. With time increasing from 5 to 36 h at 300 °C, 100 MPa, and 0.002 mol/L initial solution, the $OsO_2$ crystals varied from irregular nanoparticles to nanospheres and grew from 40–150 nm to 150–450 nm in size. As the temperature increases, the $OsO_2$ nanosphere crystals became more uniform and larger due to the promotion of the hydrolysis of the Os–chloride complex, attributed to high temperature. High initial solution concentrations precipitate more $OsO_2$ precipitates and promote the growth of the $OsO_2$ crystals. These conclusions suggest that the growth of the $OsO_2$ nanosphere crystals in the aqueous solution is complex and is mainly controlled by the nucleation and aggregate growth pattern before the hydrolysis reaction reaches the equilibrium but is dominated by the Ostwald ripening during the whole process of growth, especially after the equilibrium. According to the high surface area and good electrode conductivity, The $OsO_2$ nanosphere crystals may have great application potential as microbial reagents, catalysts, electrode conductivity materials, and in other fields. Based on the relationship between the morphology and size of $OsO_2$ crystals and the experimental conditions, this study considered that the $OsO_2$ nanosphere crystals could be a typomorphic mineral for hydrothermal vent systems and provide an indirect understanding of the transport and enrichment of Os in the form of Os–chloride complexes in seafloor hydrothermal systems.

**Author Contributions:** Conceptualization, H.Y. and X.D.; methodology, Z.L.; software, J.D.; validation, H.Y.; formal analysis, H.Y.; investigation, X.D.; resources, H.Y.; data curation, J.D.; writing—original draft preparation, H.Y.; writing—review and editing, X.D.; visualization, H.Y.; supervision, X.D.; project administration, X.D. All authors have read and agreed to the published version of the manuscript.

**Funding:** This study was funded by the National Natural Science Foundation of China, grant number (41730423) and the Strategic Priority Research Program of the Chinese Academy of Sciences, grant number (XDB42000000).

**Data Availability Statement:** Data sharing is not applicable to this article.

**Acknowledgments:** We thank the reviewers for their constructive suggestions. This work is a contribution No. IS-3237 from GIGCAS.

**Conflicts of Interest:** The authors declare no conflict of interest.

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
