# Peer review of "Crystal Growth of Osmium(IV) Dioxide in Chlorine-Bearing Hydrothermal Fluids"

_minerals, doi:10.3390/min12091092_

Round 1
Reviewer 1 Report
Minor:
Title: “Crystal Growth of Osmium(IV) Dioxide in Chlorine-bearing Hydrothermal Fluids”. Here “Osmium(IV) Dioxide” should be replaced by “Osmium(IV) oxide” for proper nomenclature. Same modification recommended for page 2, line 47.
Line 108: (Chemical equation) Change the double-headed arrow to single-headed arrows to represent reversibility. Substitute the down-ward arrow in the product side with “(s)” and include the phase of the reactants and products as “(aq)”. Remove the period at the end of the equation. Images in figure 3 and figure 5 are at different magnification levels, provide images with a common magnification so that visual comparison is easier.
Major: This article discusses crystal growth. Size of the particle may be large due to aggregation of crystals and adopt spherical shapes of different dimensions. For that reason, X-ray diffraction is necessary. This would allow the authors to calculate the crystal size using Scherrer equation.
Author Response
Dear Reviewer,
We sincerely thank the constructive comments, which are very helpful for us to improve our manuscript. The review comments are considered carefully and addressed in detail below. Corresponding modifications are made in the manuscript whenever necessary (marked in red).
Detailed response:
Reviewers' comments are marked as “C” and italicized text, and responses are marked as “R” in the following text:
C1:Title: “Crystal Growth of Osmium(IV) Dioxide in Chlorine-bearing Hydrothermal Fluids”. Here “Osmium(IV) Dioxide” should be replaced by “Osmium(IV) oxide” for proper nomenclature. The same modification is recommended for page 2, line 47.
R: Thanks. It is modified.
C2:Line 108: (Chemical equation) Change the double-headed arrow to a single-headed arrow to represent reversibility. Substitute the downward arrow on the product side with “(s)” and include the phase of the reactants and products as “(aq)”. Remove the period at the end of the equation. Images in figure 3 and figure 5 are at different magnification levels, provide images with a common magnification so that visual comparison is easier.
R: Thanks. It is modified. The double-headed arrow, the down-ward arrow, and the phases of the reactants and products of the Chemical equation in Line 112 are changed according to the reviewer’s advice. The period at the end of the equation in Line 112 is removed. Most of the images in figure 4 and figure 6 have the same magnification with ~20.0 k and show clear morphology of OsO2 crystals(Figure 4b, 4d,4f, 6a, 6b). Having different magnifications from 4.0k to 70.0 k, Other images in figure 4 and figure 6 also show the clear crystal structure of OsO2 without blurring. The images with different magnification can reflect the characteristics of crystal morphology of OsO2 in parts and whole. The measurement of OsO2 crystals in size strictly adopted the average value of multiple measurements to reduce the error. Each measurement can be carried out with high-precision software. Thus, the measurement result will not be affected by the different magnification of the image. Although the SEM photos with the same magnification are better, we believe that it is feasible to present the crystal structure of OsO2 crystals with the images at different magnifications in this paper. It has no significant effect on the morphology observation and measurement of the OsO2 crystals.
C3: This article discusses crystal growth. The size of the particle may be largely due to the aggregation of crystals and adopting spherical shapes of different dimensions. For that reason, X-ray diffraction is necessary. This would allow the authors to calculate the crystal size using the Scherrer equation.
R: Thanks. As shown in Figure 3. The XRD showed that the experimental products could be the OsO2 crystals.
Reviewer 2 Report
This study reported crystal growth of OsO2 through the hydrolysis experiments of K2OsCl6 to decipher the growth mechanism of OsO2 and the transport and enrichment of Os.
Reviewer asks the following questions.
1. Can you say in Figures 1-3 that there is no change of the structure of nanosphere crystals? XRD measurement may be also useful to confirm that.
2. Figure 4; The crystal mechanism is discussed for each time zone in detail. However, the data is dispersed largely.
3. Figure7; How is the concentration change of the solution during the Ostwald ripening.
4. Magnification of SEM Photo is different between each conditions. So, the discrimination is difficult.
This study reported crystal growth of OsO2 through the hydrolysis experiments of K2OsCl6 to decipher the growth mechanism of OsO2 and the transport and enrichment of Os.
Reviewer asks the following questions.
1. Can you say in Figures 1-3 that there is no change of the structure of nanosphere crystals? XRD measurement may be also useful to confirm that.
2. Figure 4; The crystal mechanism is discussed for each time zone in detail. However, the data is dispersed largely.
3. Figure7; How is the concentration of the solution during the Ostwald ripening.
4. Magnification of SEM Photo is different between each conditions. So, the discrimination is difficult.
v
Author Response
Dear Reviewer,
We sincerely thank the constructive comments, which are very helpful for us to improve our manuscript. The review comments are considered carefully and addressed in detail below. Corresponding modifications are made in the manuscript whenever necessary (marked in red).
Best regards,
Detailed response:
Reviewers' comments are marked as “C” and italicized text, and responses are marked as “R” in the following text:
C1:Can you say in Figures 1-3 that there is no change in the structure of nanosphere crystals? XRD measurement may be also useful to confirm that.
R: Thanks. The XRD measurement have been shown in Figure 3. The XRD showed that the experimental products are the OsO2 crystals. Figures 1-3 all showed that the structure of the OsO2 crystals is not significant, but mainly the change of particle size of crystals.
C2:Figure 4; The crystal mechanism is discussed for each time zone in detail. However, the data is dispersed largely.
R: Thanks. According to previous studies, the hydrolysis of metal complexes can reach equilibrium within a few hours, such as a dozen hours [1-3]. Based on this result, five runtimes containing 1, 5, 12, 24, and 36 hours were adopted in this study to define the equilibrium time of hydrolysis for the Os-Cl complex. The time series changed from a few hours to tens of hours and were sufficient to determine the equilibrium time. However, the SEM micrograph of OsO2 at 1 hour showed unnucleated and extremely dispersed particles and had little effect on the analysis of the OsO2 crystal structure (Figure.S 1). Therefore, the SEM micrograph of OsO2 at 1 hour was not mentioned in this paper. Other runtimes including 5, 12, 24, and 36 hours in this study were dispersed but enough to determine the equilibrium time of hydrolysis for the Os-Cl complex. The size of the OsO2 crystals at 5-36 hours could reflect the crystal mechanism. More intensive runtimes data may be better but makes little difference to the results. Thus, the data from 5, 12, 24, and 36 hours was dispersed but enough to discuss the crystal mechanism of OsO2.
C3: Figure7; How is the concentration change of the solution during the Ostwald ripening?
R: Thanks. The Ostwald ripening can occur during the whole hydrolysis of the Os-Cl complex but dominate after the hydrolysis gets equilibrium. Before the hydrolysis reaches equilibrium within 24 hours, reaction (1) can occur forward, the Os concentration of the reactive solution can decrease and the HCl and Cl- concentrations of the reactive solution can increase. At this time, a little effect of the Ostwald ripening on the growth of the OsO2 crystals, and the growth of the OsO2 crystals could be mainly controlled by the nucleation and aggregate growth. When the hydrolysis reaches hydrolysis, reaction (1) can be in dynamic equilibrium, and the Os, HCl, and Cl- of the reactive solution can keep stable. The specific concentration change of the solution was shown in Figure.S 2 (unpublished). The results showed that the Os concentration of reactive solution changes from hundreds of ppm to several ppm with time increasing, providing evidence for the concentration change of reactive solution.
C4: Magnification of SEM Photo is different between each condition. So, discrimination is difficult.
R: Thanks. Although the magnification of SEM photo varies and is between 4.0 k and 70.0 k, most of the images in figures 4 and 6 have the same magnification with ~20.0 k and show clear morphology of OsO2 crystals(Figure 4b, 4d,4f, 6a, 6b). With different magnifications from 4.0 k to 70.0 k, the images in figures 4 and 6 also show the clear crystal structure of OsO2 without blurring. Although the SEM photos with the same magnification are better, the images with different magnification can also reflect the characteristics of crystal morphology of OsO2 in parts and whole. The measurement of OsO2 crystals in size strictly adopted the average value of multiple measurements to reduce the error. Each measurement can be carried out with high-precision software. Thus, the measurement result will not be affected by the different magnification of the image. Therefore, we believe that it is feasible to present the crystal structure of OsO2 crystals with the images at different magnifications in this paper. It has no significant effect on the morphology discrimination and measurement for the OsO2 crystals.
References
- Ding, X.; He, J.J.; Liu, Z.Y. Experimental Studies on Crystal Growth of Anatase under Hydrothermal Conditions. Earth Science 2018, 43, 1763-1772.
- Yan, H.B.; Sun, W.D.; Liu, J.F.; Tu, X.L.; Ding, X. Thermodynamic properties of ruthenium (IV) chloride complex and the transport of ruthenium in magmatic-hydrothermal fluids. Ore Geology Reviews 2021, 131, 104043, doi:10.1016/j.oregeorev.2021.104043.
- He, J.J.; Ding, X.; Wang, Y.R.; Sun, W.D. The effects of precipitation-aging-re-dissolution and pressure on hydrolysis of fluorine-rich titanium complexes in hydrothermal fluids and its geological implications. Acta Petrol Sin 2015, 31, 1870-1878 (in Chinese with English abstract).

Reviewer 3 Report
The Manuscript should be published pending some minor revisions. Figures are the most important to be improved. Also, please improve results and discussion.
Review of Crystal Growth of Osmium (IV) Dioxide in Chlorine-bearing Hydrothermal Fluids by Yan et al.
General comments:
1. The figures depicting OsO2 crystal growth (Fig. 3 and 5) – the scale bar is between 1 - 10 micron. Why not get higher resolution SEM images? The previous publications have been able to present either high resolution images accompanied or the diagrams representing crystalline structure and bonding. That is not technically difficult and should be included in the paper.
2. I also did not see an adequate explanation why the condensed material that crystallize from the fluid solution is composed of “spherules” and not of typical crystallite structures (OsO2 belongs to Tetragonal P4â‚‚/mnm space group as far as I know). So, the images and what they represent here are of concern as they do not show sufficient resolution of “crystalized” material.
3. Example of Figure formatting: Fig. 4 Mark hydrolysis growth and Ostwald ripening. What are intermediate stages? Please offer a more detailed explanation. In other plots, the y-axis should contain numerical values (it cannot be arbitrary). The specialist as well as general audiences need to know intensity and counts in Fig. 1 and 2.
4. What is the time spent at each temperature in Fig. 6? Needs to be specified in the figure and text and discussed in more detail.
5. What is the activation energy for each stage of grain growth /reaction? Please specify. How does activation energy (assuming typical grain growth law) affect the rate of grain growth? This must be clearly stated in the paper that deals with the specific topic of grain growth.
6. In terms of revising the language in the text: For example, line 15 cannot say “deciphering”. That is inappropriate for a scientific paper and should be revised to more appropriate language with better scientific meaning.
7. Line 16: Time series – I did not see any time series. Either it is a typo or the authors have poor understanding what time series implies in scientific terms.
8. Line 36: “Thereinto” - please change, reword
9. Line 100: the word “paved” is inappropriate. I know what you did, and you need to find an appropriate meaning to describe it – “paved” is not it.
10. There are many more examples throughout the text, but it is not my job to do this type of editing.
11. Also, the authors should discuss more comprehensively the industrial implications as that is also a very important part of their work.
Otherwise, the manuscript is of sufficient scientific soundness (within given discipline) that it can be published.
Author Response
Dear Reviewer,
We sincerely thank the constructive comments, which are very helpful for us to improve our manuscript. The review comments are considered carefully and addressed in detail below. Corresponding modifications are made in the manuscript whenever necessary (marked in red).
Best regards,
Detailed response:
Reviewers' comments are marked as “C” and italicized text, and responses are marked as “R” in the following text:
C1: The figures depicting OsO2 crystal growth (Fig. 3 and 5) – the scale bar is between 1 - 10 micron. Why not get higher resolution SEM images? The previous publications have been able to present either high-resolution images accompanied or diagrams representing crystalline structure and bonding. That is not technically difficult and should be included in the paper.
R: Thanks. Although the scale bar of SEM photo varies and is between 1 - 10 microns, most of the images in figures 4 and 6 have the same scale bar with ~2 microns and show clear morphology of OsO2 crystals (Figure 4b, 4d,4f, 6a, 6b). The other SEM images with different magnifications in figures 4 and 6 also show the clear crystal structure of OsO2 without blurring. Although the SEM photos with the higher resolution are better for the observation of the OsO2 crystals, the images with magnification from 4.0 k to 70.0 k can also reflect the characteristics of crystal morphology of OsO2 in parts and whole. The measurement of OsO2 crystals in size strictly adopted the average value of multiple measurements to reduce the error. Each measurement can be carried out with high-precision software. Thus, the measurement result will not be affected by the low-resolution SEM images. Therefore, we believe that the SEM images with the scale bar changing from 1 to 10 micron are feasible to present the crystal structure of OsO2 crystals in this paper. The higher resolution SEM images are better but unnecessary.
C2: I also did not see an adequate explanation why the condensed material that crystallize from the fluid solution is composed of “spherules” and not of typical crystallite structures (OsO2 belongs to Tetragonal P4â‚‚/mnm space group as far as I know). So, the images and what they represent here are of concern as they do not show sufficient resolution of “crystalized” material.
R: Thanks. As mentioned in section 3.5, the previous study just synthesized the rutile-type OsO2 crystal by the chemical-vapor-transport (CVT) method at high temperatures (800-940 oC). However, the OsO2 nanosphere crystals for the first time were synthesized by the hydrolysis of the Os chloride complex in this study. Therefore, the occurrence mechanism of the synthesized OsO2 is unclear and related to the formation environments. Based on the previous research and this study, the formation of the OsO2 nanosphere crystals may be attributed to two reasons: 1) the temperature and pressure. The rutile-type OsO2 crystals are formed under the conditions of high temperature (800-940 oC) and saturated vapor pressure, but the synthesis of the OsO2 nanosphere crystals is closely related to a lower temperature (150-550 oC) and higher pressure (100 MPa). Higher temperature and lower pressure may be conducive to the complete growth of the crystal structure (Tetragonal P4â‚‚/mnm space group), promoting the rapid formation of the rutile-type OsO2 crystals. However, lower temperature and higher pressure may not allow the OsO2 crystal to completely grow according to the lattice structure[1]. 2) the occurrence environment. Cl and F can inhibit the crystal growth and change the crystal morphology, such as the fluorine in the fluids can reduce the surface energy and promote the growth of the (001) facets of ZrO2, resulting in the formation of zirconia micro-nanoflake with high (001) facets [2,3]. In this study, the inhibition of Cl in fluids could cause the OsO2 crystal morphology to change and cannot grow according to the original lattice structure. Moreover, the low Os saturation also plays an effect on the formation of OsO2 crystals in form of a nanosphere. Therefore, the occurrence of the OsO2 nanosphere crystals in this study may be attributed to a combination of factors, such as temperature, pressure, and fluid composition. The morphology of the OsO2 synthesized crystals can be observed in Figures 4 and 6, showing sufficient resolution for the observation.
C3:Example of Figure formatting: Fig. 4 Mark hydrolysis growth and Ostwald ripening. What are intermediate stages? Please offer a more detailed explanation. In other plots, the y-axis should contain numerical values (it cannot be arbitrary). The specialist as well as general audiences need to know the intensity and counts in Fig. 1 and 2.
R: Thanks. As mentioned in section 3.4, the nucleation and aggregate growth and the Ostwald ripening can occur during the whole process of hydrolysis but play different roles at different stages. Before the hydrolysis reaction reaches equilibrium, reaction (1) moves to the right, and a large amount of OsO2 precipitates from the fluids to quickly aggregate and nucleate. During this process, the nucleation and aggregate growth control the growth of OsO2 crystals. The Ostwald ripening also occurs and plays a non-dominant role, but can be gradually enhanced with the continuous hydrolysis reaction. On the contrary, the nucleation and aggregate growth can be less and less significant for the growth of OsO2 crystals with the increase of the runtime. When the hydrolysis is close to or reaches the equilibrium, the amount of OsO2 precipitates can remain the same, the Ostwald ripening instead of the nucleation and aggregate growth can dominate the growth of OsO2 crystals and promote a more complete morphology of the OsO2 crystals. Therefore, the nucleation and aggregate growth occur mainly before equilibrium but are insignificant after equilibrium. The Ostwald ripening can occur throughout the whole process of hydrolysis and could be the dominant factor for the growth of OsO2 crystals in the early stage, but gradually dominating in the late stage, especially after the hydrolysis reaches equilibrium. The y-axis of figures. 1 and 2 is modified.
C4: What is the time spent at each temperature in Fig. 6? Needs to be specified in the figure and text and discussed in more detail.
R: Thanks. As mentioned in the legend of the figure. 7, the runtime of all experiments at 150-550 oC in the figure. 7 is 24 hours to eliminate the effect of time. The result from the figure. 7 shows the size (the median) of OsO2 crystals as a whole shows gradually increases from 160-180 to 215-240 nm with temperatures elevated from 150 to 550 oC, indicating the promotion of elevated temperature for the OsO2 crystals in size. A detailed discussion is in section 3.2.
C5: What is the activation energy for each stage of grain growth /reaction? Please specify. How does activation energy (assuming typical grain growth law) affect the rate of grain growth? This must be clearly stated in the paper that deals with the specific topic of grain growth.
R: Thanks for the reviewer’s advice. The activation energy for each stage of the reaction could help us understand the OsO2 crystal growth better. The activation energy should be calculated based on the rate of the hydrolysis reaction. However, the rate of hydrolysis reaction (1) does not equal the rate of grain growth, and the relevant data are not mentioned in this paper but will be detailed discussion in the following paper (unpublished). Therefore, at present, the calculated activation energy of the hydrolysis reaction of the Os-Cl complex based on the rate of grain growth is not accurate in this paper. Moreover, the activation energy of the hydrolysis reaction could play a limited role in the grain growth based on the unpublished data. The change of Gibbs free energy of the hydrolysis reaction of the Os-Cl complex is between 273 and 106 kJ/mol at 150-550 oC and the hydrolysis rate of the Os-Cl complex is up to more than 90 % at different temperatures (unpublished). The results from Figure.S 2 also showed that the hydrolysis reaction could occur rapidly and the hydrolysis rate of the Os-Cl complex is up to more than 95 % at 300 oC. This means that the rate of reaction (1) can be faster as the temperature is elevated and the hydrolysis reaction can occur rapidly even at low temperatures (150 oC), indicating the low activation energy of the hydrolysis reaction (1). At higher temperatures, the hydrolysis reaction can occur rapidly, resulting in the rapid growth of the OsO2 crystals. Thus, the activation energy has no significant effect on the growth of OsO2 grains with elevated temperatures.
C6: In terms of revising the language in the text: For example, line 15 cannot say “deciphering”. That is inappropriate for a scientific paper and should be revised to more appropriate language with better scientific meaning.
R: Thanks. It is modified.
C7: Line 16: Time series – I did not see any time series. Either it is a typo or the authors have a poor understanding of what time series implies in scientific terms.
R: Thanks. As shown in section 3.1, a series of experiments were carried out from 5, 12, 24, and 36 hours at 300 °C, 100 MPa, and 0.002 mol/L initial solutions. These experiments were performed at different runtimes and can be called time-series experiments.
C8: Line 36: “Thereinto” - please change, reword
R: Thanks. It is modified.
C9: Line 100: the word “paved” is inappropriate. I know what you did, and you need to find an appropriate meaning to describe it – “paved” is not it.
R: Thanks. It is modified.
C10: There are many more examples throughout the text, but it is not my job to do this type of editing.
R: Thanks. This paper can be modified in detail.
C11: Also, the authors should discuss more comprehensively the industrial implications as that is also a very important part of their work.
R: Thanks. As shown in section 3.4, the OsO2 nanosphere crystals are obtained for the first time and likely have great application potentials on microbial reagents, catalysts, gas fixing agents, and electrode conductivity materials because of their high surface area and good electrode conductivity[4-7]. Moreover, according to the synthesis conditions and crystal morphology of grains, the growth of OsO2 nanosphere crystals may be used as the hypomorphic mineral to trace the hydrothermal conditions and fluids properties. A detailed discussion is in section 3.5.
References
- Gao, B.; Nakano, S.; Kakimoto, K. The impact of pressure and temperature on growth rate and layer uniformity in the sublimation growth of A1N crystals. J Cryst Growth 2012, 338, 69-74.
- Yang, H.G.; Sun, C.H.; Qiao, S.Z.; Zou, J.; Liu, G.; Smith; Campbell, S.; Cheng, H.M.; Lu, G.Q.J.N. Anatase TiO2 single crystals with a large percentage of reactive facets. Nature 2008, 453, 638-642.
- Yan, H.B.; Di, J.; Li, J.; Liu, Z.; Liu, J.; Ding, X. Synthesis of Zirconia Micro-Nanoflakes with Highly Exposed (001) Facets and Their Crystal Growth. Crystals 2021, 11, 1-11, doi:https://doi.org/10.3390/cryst11080871.
- Graebner, J.E.; Greiner, E.S.; Ryden, W.D. Magnetothermal oscillations in RuO2, OsO2, and IrO2. Phys Rev B 1976, 13.
- Mattheiss, L.F. Electronic structure of RuO2, OsO2, and IrO2. Phys Rev B 1976, 13, 2433-2450.
- Hayakawa, Y.; Kohiki, S.; Arai, M.; Yoshikawa, H.; Fukushima, S.; Wagatsuma, K.; Oku, M.; Shoji, F. Electronic structure and electrical properties of amorphous OsO2. Phys Rev B 1999, 59, 11125.
- Heidari, A.; Hotz, M.; MacDonald, N.; Peterson, V.; Caissutti, A.; Besana, E.; Esposito, J.; Schmitt, K.; Chan, L.-Y.; Sherwood, F.; et al. Osmium Dioxide (OsO2) and Osmium Tetroxide (OsO4) Smart Nano Particles, Nano Capsules and Nanoclusters Influence, Impression and Efficacy in Cancer Prevention, Prognosis, Diagnosis, Imaging, Screening, Treatment and Management under Synchrotron and Synchrocyclotron Radiations. International Journal of Physics 2022, 10, 1-22.